# Peroxisome Proliferator-Activated Receptor α Has a Protective Effect on Fatty Liver Caused by Excessive Sucrose Intake

**DOI:** 10.3390/biomedicines10092199

**Published:** 2022-09-06

**Authors:** Tomomi Yamazaki, Megumi Ihato

**Affiliations:** Department of Nutrition and Metabolism, National Institute of Health and Nutrition, National Institutes of Biomedical Innovation, Health and Nutrition, 1-23-1 Toyama, Shinjuku-ku, Tokyo 162-8636, Japan

**Keywords:** NAFLD, PPARα, PPARγ, SREBP-1c, sucrose

## Abstract

Sterol regulatory element binding protein (SREBP)-1c is a transcription factor that regulates lipid synthesis from glucose in the liver. It is activated by sucrose, which activates the fatty acid synthesis pathway. On the other hand, peroxisome proliferator-activated receptor (PPAR) α regulates the transcription of several genes encoding enzymes involved in fatty acid β-oxidation in the liver. To evaluate the beneficial effects of PPARα on fatty liver caused by excessive sucrose intake, we investigated the molecular mechanisms related to the development of fatty liver in PPARα-deficient mice that were fed a high-sucrose diet (Suc). The SREBP-1c target gene expression was increased by sucrose intake, leading to the development of fatty liver. Furthermore, PPARα^−/−^ mice developed severe fatty liver. Male and female PPARα^−/−^ mice fed Suc showed 3.7- and 3.1-fold higher liver fat content than Suc-fed male and female wild-type mice, respectively. Thus, PPARα may work to prevent the development of fatty liver caused by excessive sucrose intake. Liver TG accumulation differed between male and female PPARα^−/−^ mice. A possible explanation is that male mice show the increased expression of *Pparγ*, which usually contributes to triglyceride synthesis in the liver, to compensate for *Pparα* deficiency. In contrast, female wild-type mice inherently have low *Pparα* levels. Thus, *Pparα* deficiency has less pronounced effects in female mice. A diet that activates PPARα may be effective for preventing the development of fatty liver due to excessive sucrose intake.

## 1. Introduction

In Western societies, there has been a dramatic increase in the prevalence of obesity in association with increased fat or carbohydrate intake. As a result of obesity, there is a new epidemic of hepatic steatosis and non-alcoholic fatty liver disease (NAFLD) [1]. Non-alcoholic steatohepatitis (NASH) is an advanced form of NAFLD, characterized by macrovesicular steatosis and parenchymal inflammation [2]. The prevalence of NAFLD in the general population is 10–24%, while NASH affects approximately 3% of the lean population and almost half of the morbidly obese population [3]. The term NAFLD was recently renamed metabolic dysfunction associated fatty liver disease (MAFLD) to better reflect the patient heterogeneity and pathogenesis [4]. The accumulation of hepatic lipids in NAFLD patients is caused by plasma nonesterified fatty acids (NEFAs) from adipose tissue, fatty acids produced in the liver by de novo lipogenesis, and fatty acids derived from dietary sources, which flow into the liver via NEFA efflux from chylomicron lipolysis and the hepatic uptake of chylomicron remnants. The analysis of multiple stable isotopes in NAFLD patients revealed that 59% of liver triglycerides (TG) are derived from NEFA, 26% from de novo lipogenesis, and 15% from dietary fatty acids [5]. Approximately one-quarter of liver fatty acids are generated from de novo 2-carbon precursors derived from glucose, fructose, and amino acids. Thus, in addition to dietary fat, and sugars, such as sucrose, glucose, and fructose, also significantly contribute to the liver fatty acid storage pool.

Sterol regulatory element (SRE)-binding protein (SREBP)-1c is a transcription factor that controls the synthesis of lipids from glucose in the liver, and one of the members of the basic helix-loop-helix-leucine zipper (bHLH-Zip) family of transcription factors [6]. Three SREBP isoforms, named SREBP-1a, SREBP-1c, and SREBP-2, are encoded in the mammalian genome [7]. SREBP-1c plays a stronger role in regulating fatty acid synthesis in the liver [8]. In contrast, SREBP-1a is a potent activator of genes mediating the synthesis of cholesterol, fatty acids, TG, and phospholipids; however, it is less abundant in the liver than SREBP-1c [9,10,11]. SREBP-2 selectively activates the gene expression of cholesterol biosynthetic enzymes and transporters [8]. Stimulation of the expression of lipogenic genes by insulin is mediated by SREBP-1c, which leads to the activation of all genes required for the synthesis of fatty acids and the first enzyme in the synthesis of TG [5]. SREBP-1c is synthesized as a precursor protein and is retained in an inactive form in the endoplasmic reticulum, where it is bound to two other proteins: insulin-induced gene (INSIG) and SREBP cleavage-activating protein (SCAP) [6,12]. The activation of SREBP-1c requires dissociation from INSIG proteins and translocation of the SREBP/SCAP complex to the Golgi complex, where SREBP-1c undergoes a two-step proteolytic process by two proteases—site 1 protease and site 2 protease—to release the transcriptionally active *N*-terminal fragment for nuclear entry [6,13,14,15]. The excessive intake of carbohydrates, especially sucrose and fructose, leads to the transcriptional activation of glycolytic and lipogenic genes, and this process is regulated by complex regulatory processes that require the coordinated action of both insulin and glucose [16,17,18,19]. SREBP-1c is one of the major transcription factors responsible for the coordinated induction of glycolytic and lipid metabolism genes [6,20]. The other is the glucose-sensing transcription factor carbohydrate-responsive element-binding protein (ChREBP). Glucose activates genes with carbohydrate responsive elements (ChoRE) [21]. ChREBP has ChoRE and is also a member of the bHLH-Zip transcription factor family [22]. ChREBP mRNA is widely expressed in the liver, adipose tissue, and elsewhere [22,23]. SREBP-1c and ChREBP interdependently regulate the expression of glycolytic and lipogenic mRNAs, indicating that both SREBP-1c and ChREBP are required for sucrose intake to induce the accumulation of hepatic TG [24]. These studies further elucidate the molecular events that contribute to the development of fructose overdose-induced hepatic lipidosis, which is prevalent in Western societies [25,26].

Peroxisome proliferator-activated receptor (PPAR) α is one of the isotypes of PPAR and is highly expressed in the liver. PPARs regulate many metabolic pathways upon activation by endogenous ligands (e.g., fatty acids or synthetic agonists) after binding to PPAR responsive element (PPRE) in the promoter region of genes [27]. PPARα controls the transcription of several genes encoding enzymes involved in fatty acid uptake, binding, and activation, mitochondrial fatty acid β-oxidation, and hydrolysis of plasma TG, which contain a PPRE in their promoter region [28]. The transcription factor PPARγ is another isotype of PPAR which is mainly involved in the development of fatty liver in association with high-fat (HF) diet feeding [16,17,29,30]. HF diets have been reported to activate not only PPARγ, but also PPARα target genes by increasing PPARα mRNA and plasma free fatty acid (FFA) concentrations [31]. An HF diet increases the amounts of fatty acids that reach the liver, and the requirement for fatty acid oxidation is concomitantly increased. Despite the upregulation of PPARα and numerous PPARα target genes involved in fatty acid oxidation, an HF diet causes the development of fatty liver. This suggests that the upregulation of PPARα is not sufficient for the excess load of fatty acids to be efficiently catabolized. The deletion of PPARα in male mice resulted in more pronounced hepatic accumulation of TG during HF diet feeding, suggesting that PPARα protects against lipid overload [31]. Moreover, it has also been reported that an HF diet and hepatic *Ppar**α* deficiency independently increase liver TG levels [31,32].

A high-sucrose diet causes liver steatosis depending on de novo lipogenesis, promoting ectopic fat deposition in the liver. Previously, we reported that the activation of PPARα prevents fatty liver development caused by sucrose intake [16]. Therefore, to investigate the function of PPARα during the high-sucrose diet-induced development of fatty liver, we examined the effects of a high-sucrose diet on the liver in male and female PPARα-deficient mice.

## 2. Materials and Methods

### 2.1. Animals

PPARα-null mice were obtained from The Jackson Laboratory (Bar Harbor, ME, USA). Wild type (WT) 129S1/SvImJ mice were used as controls (The Jackson Laboratory). They were fed a standard laboratory diet (CE2) from CLEA Japan, Inc, (Tokyo, Japan) until use for research. Mice were maintained under a controlled environment at 22 °C in a 12-h light (07:00–19:00 h)/12-h dark (19:00–07:00 h) cycle with *ad libitum* access to chow and water. The care of the mice was in accordance with the guidelines of the National Institutes of Health’s Guide for the Care and Use of Laboratory Animals. The National Institutes of Biomedical Innovation, Health and Nutrition, Japan, reviewed and approved all animal procedures (Approval no. DS27-52R3).

### 2.2. Diet

Mice (age: 35–40 weeks) received a starch diet (St) (70 en% starch, control) or high-sucrose diet (Suc) (52.5 en% sucrose) groups. The detailed composition of each experimental diet is listed in Table 1. Diets were prepared as described in our previous studies [33]. Butter was purchased from Snow Brand Milk Corp. (Hokkaido, Japan). Safflower oil was purchased from Benibana Food (Tokyo, Japan). Other dietary components were purchased from Oriental Yeast Co., Ltd. (Tokyo, Japan). To estimate daily food intake, the food weight of each day was subtracted from the initial food weight of the previous day. The mean food intake over the entire experimental period in the two groups of mice was calculated using these data. The diets were provided for 4 weeks.

### 2.3. Serum Chemistry

Blood was obtained from the mice, and serum glucose was measured with a glucometer (FreeStyle Freedom Lite, NIPRO, Osaka, Japan). Serum levels of NEFA, TG and total cholesterol (TC) were measured by enzymatic colorimetry with NEFA C [34], TG E [35] and TC E [36] test kits (Wako Pure Chemical Industries, Ltd., Osaka, Japan), respectively.

### 2.4. Glucose and Insulin Tests

For the glucose tolerance test (GTT), mice were fasted overnight and then D-glucose (1 g/kg body weight) was administered orally. For the insulin tolerance test (ITT), food was removed from mice 4 h prior to the experiment and recombinant insulin (0.75 U/kg, i.p.; Humulin R, Eli Lilly Japan K.K., Kobe, Japan) was administered. GTT and ITT performed at the beginning of the fourth week.

### 2.5. Hepatic Histology

Mouse livers were fixed in 4% neutral-buffered formalin, embedded in paraffin, cut into sections, and stained with hematoxylin and eosin.

### 2.6. Quantitative Real-Time PCR

At the end of the experiment, mice were sacrificed by cervical dislocation and the liver was isolated. RNA was extracted with TRIzol Reagent (Molecular Research Center, Inc., Cincinnati, OH, USA) in accordance with the manufacturer’s instructions. RNA was isolated and quantified with a NanoDrop ND-2000 spectrophotometer (Thermo Fisher Scientific, Waltham, MA, USA). Total RNA was reverse transcribed, and quantitative real-time PCR was performed as described previously [37]. The primers for quantitative real-time PCR are listed in Table 2.

### 2.7. Statistical Analysis

Values are shown as the mean and standard error. A two-way ANOVA was used to examine the two main effects of strain and diet and their interaction (IBM SPSS Statistics 23). When we found a significant interaction, we performed a Test of Simple Effects with SPSS using the Estimated Marginal Means option. *p* values of <0.05 were considered to indicate statistical significance.

## 3. Results

### 3.1. Body Weight and Tissue Weights

The results for body weight and the individual tissue weights of PPARα^−/−^ and WT mice on control or Suc-fed mice are shown in Table 3. The energy intake was lower in male mice in the Suc-fed group and lower in female mice in the PPARα^−/−^ and Suc-fed groups. Thus, the Suc-fed group had a lower food intake, but pair-feeding was not performed in this study to eliminate the effect of increased NEFA in blood due to fasting. In male mice, no differences in body weight were observed between strains or food types. However, the liver weight was significantly greater in PPARα^−/−^ mice (*p* < 0.001) and was also significantly increased by sucrose intake (*p* = 0.001). In PPARα^−/−^ mice, the white adipose tissue (WAT) weight of retroperitoneal WAT (RetroWAT) and subcutaneous WAT (SubWAT) was significantly greater (RetroWAT; *p* = 0.005, SubWAT; *p* = 0.012), while RetroWAT and mesenteric WAT (MesWAT) were significantly increased by sucrose intake (RetroWAT; *p* = 0.028, MesWAT; *p* = 0.012). The brown adipose tissue (BAT) weight was significantly lower in PPARα^−/−^ mice (*p* < 0.001). There were no differences in epididymal WAT (EpiWAT) or muscle tissue weight. In female mice, as in males, the body weight did not differ according to strain or food type; however, the liver weight was significantly greater in PPARα^−/−^ mice (*p* = 0.004) and was also significantly increased by sucrose intake (*p* = 0.026). As for the WAT weight, the weights of periuterine WAT (PeriWAT) and MesWAT were significantly greater in PPARα^−/−^ mice (PeriWAT; *p* < 0.001, MesWAT; *p* = 0.005), and sucrose intake showed no effect on WAT weight. The gastrocnemius weight was significantly lower in PPARα^−/−^ mice (*p* = 0.016).

### 3.2. Serum Analysis

The results of the serum analysis are shown in Table 4. Blood glucose levels were significantly lower in both male and female PPARα^−/−^ mice (male; *p* < 0.001, female; *p* = 0.003), but did not differ according to diet. In male mice, the TG and TC concentrations were increased by sucrose intake (TG; *p* = 0.015, TC; *p* = 0.034). NEFA concentrations were significantly higher in PPARα^−/−^ mice (*p* = 0.039) and were increased by sucrose intake (*p* < 0.001). In female mice, TG and NEFA concentrations were significantly higher in PPARα^−/−^ mice (TG; *p* = 0.041, NEFA; *p* < 0.001). In female mice, no differences were observed in any of the parameters due to sucrose intake.

### 3.3. Hepatic Lipid Analysis

The results of the liver lipid analysis are shown in Figure 1. In male mice, PPARα^−/−^ mice showed higher liver TG levels in comparison to WT mice, and Suc feeding increased the liver TG levels (Figure 1A). In female mice, PPARα^−/−^ mice showed higher liver TG levels in comparison to WT mice in both the St and Suc intake groups. Moreover, Suc-fed PPARα^−/−^ mice showed extremely increased liver TG levels, which were higher in comparison to St-fed PPARα^−/−^ mice. The liver TC levels of male PPARα^−/−^ mice were lower in comparison to WT mice in both the St and Suc intake groups (Figure 1B). In female mice, the TC content differed by strain, with PPARα^−/−^ mice having lower liver TC content. Hematoxylin-eosin staining of liver tissue sections also showed these results (Figure 2). In PPARα^−/−^ male mice, both St- and Suc-fed mice showed higher fat deposition in comparison to WT mice (Figure 2A–D). In female mice, the most significant fat deposition was observed in Suc-fed PPARα^−/−^ mice (Figure 2E–H).

### 3.4. Hepatic mRNA Expression Analysis

We analyzed the mRNA expression in the liver to investigate the mechanism underlying the accumulation of fat in the liver. The expression levels of genes involved in lipid metabolism in male mice are shown in Figure 3. We examined the mRNA levels of transcription factor SREBP-1c, which is involved in fatty acid synthesis, and its target lipogenic genes, fatty acid synthase (*Fas*), acetyl-CoA carboxylase 1 (*Acc1*), and stearoyl-CoA desaturase 1 (*Scd1*) (Figure 3A). The expression of *Srebp-1c* was markedly lower in PPAR^−/−^ mice and sucrose intake did not cause any changes; *Fas* and *Acc1* were upregulated by sucrose consumption in both WT and PPARα^−/−^ mice. The expression of *Scd1* in PPARα^−/−^ mice was lower in comparison to St-fed WT mice. Suc-fed PPARα^−/−^ mice showed the increased expression of *Scd1* in comparison to St-fed PPARα^−/−^ mice, but showed no difference from Suc-fed WT mice. The mRNA expression of PPARα, a transcription factor responsible for fatty acid oxidation, did not differ by diet in WT mice (Figure 3B). We next examined carnitine palmitoyltransferase 1 (*Cpt1*), medium-chain acyl-CoA dehydrogenase (*Mcad*), and fibroblast growth factor (*Fgf21*), target genes of PPARα. CPT1 brings fatty acids into the mitochondria for the burning of fat [38]. MACD catalyzes the initial reaction in the mitochondrial fatty acid β-oxidation cycle [39]. *Fgf21* produces FGF21, a hormone that is known to regulate lipid accumulation and glucose intolerance [40]. *Cpt1* was differentially expressed by the mouse strains, with lower expression levels in PPARα^−/−^ mice (Figure 3B); *Mcad* was upregulated by sucrose intake; in PPARα^−/−^ mice the expression of *Fgf21* was abolished by both a control diet and Suc feeding, while in WT mice it was increased by Suc feeding. We next examined the mRNA expression levels of PPARγ1 and PPARγ2, and their target genes: fatty acid translocase (*Cd36*) and adipose differentiation-related protein (*Adrp*). An HF diet caused the activation of PPARγ, which led to the upregulation of the *Cd36* mRNA expression, specifically in the liver [29]. ADRP is a lipid storage droplet-associated protein and is also upregulated by an HF diet through PPARγ activation, followed by the induction of liver steatosis [41]. *Pparγ1*, *Pparγ2* and *Cd36* were highly expressed in PPARα^−/−^ mice, and their expression levels were increased by Suc feeding in both WT and PPARα^−/−^ mice (Figure 3C). In PPARα^−/−^ mice, the expression of *Adrp* was lower in both the St- and Suc-fed groups. Next, the gene expression results for female mice are shown in Figure 4. The expression levels of *Srebp-1c* and *Fas* were lower in PPARα^−/−^ mice, and the expression of *Fas* increased with Suc feeding in both WT and PPARα^−/−^ mice (Figure 4A). In WT mice, the expression of *Acc1* was increased in Suc-fed mice. In St-fed mice, PPARα^−/−^ mice showed the increased expression of *Acc1*. Moreover, Suc-fed PPARα^−/−^ mice showed the highest expression of *Scd1*. The expression of *Pparα* did not differ by diet, as was observed in male mice (Figure 4B). The expression of *Cpt1* was increased by Suc feeding in both WT and PPARα^−/−^ mice. The *Mcad* and *Fgf21* expression levels were lower in PPARα^−/−^ mice. In particular, the reduction in the *Fgf21* expression in PPARα^−/−^ mice was as pronounced as that in male mice. In WT mice, the *Fgf21* expression was also increased by Suc feeding. The gene expression of *Pparγ1* was significantly increased in Suc-fed mice (Figure 4C). The *Pparγ2* gene expression was significantly increased in Suc-fed PPARα^−/−^ mice and WT mice. In contrast to the results in male mice, Suc-fed PPARα^−/−^ mice showed higher *Pparγ2* gene expression levels in comparison to St-fed PPARα^−/−^ mice. In addition, this expression was higher than that of Suc-fed WT mice. The *Cd36* gene expression was also differentially expressed by mouse strain, being poorly expressed in PPARα^−/−^ mice. Moreover, Suc feeding increased the expression of *Cd36* in both WT and PPARα^−/−^ mice. The *Adrp* expression was lowest in St-fed PPARα^−/−^ mice. That is, it was significantly lower than that in St-fed WT mice and Suc-fed PPARα^−/−^ mice.

### 3.5. Glucose Tolerance Test and Insulin Tolerance Test

A GTT was performed to examine glucose tolerance. The results are shown in Figure 5A. In the Suc-fed WT group, both male and female mice showed higher blood glucose levels at 0 min. The PPARα^−/−^ mice had lower blood glucose levels than the WT mice. Suc-fed WT mice had the highest blood glucose levels during the GTT, but there were no significant differences in blood glucose levels aside from those at the beginning of the test and at 30 min after the start of the test. Furthermore, there were no significant differences in blood glucose changes between the other mouse groups. The ITT results are shown in Figure 5B. No insulin resistance was observed in male or female PPARα^−/−^ mice.

## 4. Discussion

After Suc feeding, female PPARα^−/−^ mice showed an even greater increase in liver TG content. Male PPARα^−/−^ mice also showed high liver fat accumulation after Suc feeding, although male St-fed PPARα^−/−^ mice showed greater liver TG accumulation and there was no significant difference between them. These results indicate that the PPARα function is important for the prevention of fatty liver disease induced by the ingestion of sucrose.

In both male and female PPARα^−/−^ mice, the expression of *Srebp-1c* was low, but the expression levels of SREBP-1c target genes were increased by intake of Suc, similarly to WT mice. In mice on a pure 129 SV background, sucrose/fructose supplementation has been shown to induce fatty liver accompanied by the induction of the hepatic expression of *Scd1* and other lipogenic proteins in *Srebp-1c*-knockout mice, but not in *Scd1*-knockout mice [42], indicating that the importance of the expression of *Scd1* for the development of sucrose/fructose-induced fatty liver. Our results also showed that the intake of sucrose was associated with the increased expression of *Scd1* in both male and female mice. Moreover, PPARα has been shown to increase the expression of genes involved in fatty acid oxidation as well as in the de novo lipid synthesis pathway, which is mediated by SREBP-1c in the liver [43]. Thus, *Ppar**α* deficiency may suppress the activation of SREBP-1c. Indeed, the liver *F**as* expression is reported to be low in male PPARα^−/−^ mice [44]. However, the observed increase in the expression of genes involved in the de novo fatty acid synthesis pathway induced by the intake of sucrose—even in the absence of PPARα—indicates that SREBP-1c can be normally activated by proper stimulation.

Both male and female PPARα^−/−^ mice, showed higher fat accumulation in the liver in comparison to WT mice. Even though *Ppar**α* was deficient, *Ppar**γ* had a compensatory effect [31]; thus, the expression of PPARα target genes (e.g., *C**pt1* and *Mcad*) was observed in the present study. However, the increased expression of *Ppar**γ* may lead to the accumulation of TG in the liver due to the increased expression of *Ppar**γ* target genes that are involved in the synthesis of TG. In the present study, Suc feeding significantly increased the expression of *C**pt1* in male mice and *Mcad* in female mice. It was reported that a significant increase in body weight after 10 weeks of Suc feeding decreased the expression of PPARα target genes [45]. However, the expression of SREBP-1c target genes (e.g., *Fas*) was also decreased in that study [45]. In the present study, 4 weeks of Suc feeding was not associated with a significant increase in body weight, suggesting that—at a relatively early stage—fatty acid oxidation was increased in the liver to process the increased fatty acids due to increased fatty acid synthesis from sucrose. Incidentally, *Fgf21*, one of the PPARα target genes, was hardly expressed in the liver of PPARα^−/−^ mice. Perhaps, unlike *C**pt1* and *Mcad*, etc., *Ppar**γ* is not involved in its expression. FGF21 works to prevent NAFLD [46,47]. Thus, the lack of *Fgf21* expression in PPARα^−/−^ mice may be one reason for the increased accumulation of TG in the liver in PPARα^−/−^ mice. On the other hand, it has been reported that FGF21 inhibits nuclear translocation of SREBP1c and decreases the amount of mature SREBP-1c protein [48]. However, the inhibition of SREBP-1c by FGF21 was not observed in the present study.

When 18-month-old mice were fed an HF diet for 4 weeks, WT mice showed more severe insulin resistance than PPARα^−/−^ mice. This is thought to be because HF diet feeding was associated with increased fatty acid oxidation and decreased glucose utilization in WT mice, whereas PPARα^−/−^ mice showed decreased fatty acid oxidation and increased glucose utilization, and, thus, no insulin resistance [49]. The present study demonstrated that Suc feeding did not cause insulin resistance in PPARα^−/−^ mice. This is probably also due to the promotion of glucose utilization.

Marked liver TG accumulation due to Suc feeding was observed in both male and female PPARα^−/−^ mice. However, the amount of liver TG in control male PPARα^−/−^ mice was still higher than that in St-fed WT mice and—even more surprisingly—than that in Suc-fed WT mice. Since increased hepatic PPARγ2 leads to the development of fatty liver [50], the higher expression of *Pparγ2* in the liver of male PPARα^−/−^ mice, which show the compensatory expression of PPARγ, is associated with an increase in the liver TG level of male PPARα^−/−^ mice, as previously reported [31,44,51]. Similarly, *Cd36*, a PPARγ target gene, was found to be expressed at higher levels in male PPARα^−/−^ mice in the present study. The expression of *Adrp* mRNA is reported to be reduced in PPARα^−/−^ mice; however, ADRP is highly expressed at the protein level [52]. This was also demonstrated by our results. Thus, the increased accumulation of TG in the liver, which was observed in male PPARα^−/−^ mice but not in female PPARα^−/−^ mice, is likely due to the increased expression of *Pparγ2* in the liver because *Ppar**α* is not expressed in the liver of male—but not female—PPARα^−/−^ mice.

It has also been reported that female mice show lower *Ppar**α* expression levels than male mice [53]. In humans, fenofibrate, which is a PPARα agonist, works better in males than in females, indicating that sex dimorphorism in the expression of *Ppar**α* is also present in humans [54]. Therefore, in male PPARα^−/−^ mice, the loss of *Ppar**α* expression leads to the compensatory upregulation of the *Ppar**γ* and the regulation of the expression of genes with PPRE, resulting in the development of fatty liver. However, the phenotype of females does not change as much as that of males because the expression of *Ppar**α* is originally lower in females. The mechanisms by which estrogen signaling protects against hepatic steatosis include the reduction of de novo lipogenesis through the suppression of *Fas* and *Scd1* expression in the liver [55,56]. 17β-Estradiol (E2) treatment also likely promotes fatty acid oxidation in the liver, since E2 treatment induced an increase in the mRNA levels of CPT1 [57]. Thus, it is likely that E2 plays a role in suppressing the development fatty liver, which may be why the *Ppar**α* expression was observed to be lower in females. More detailed analysis may be possible by using ovariectomized female PPARα-/- mice.

The present study using Suc-fed PPARα-deficient mice revealed that PPARα plays a protective role against the induction of fatty liver development by a high sucrose diet. This was observed in both male and female mice. A diet that activates PPARα may be effective for preventing the development of fatty liver due to excessive sugar intake. We also showed that the liver accumulation of TG differed between male and female PPARα^−/−^ mice. This may be because the expression of *Ppar**γ* in PPARα^−/−^ male mice was increased in order to compensate for the loss of *Ppar**α*, while female mice were less affected by the loss of *Ppar**α* because the expression of *Ppar**α* was originally low in WT female mice.

## Figures and Tables

**Figure 1 biomedicines-10-02199-f001:**
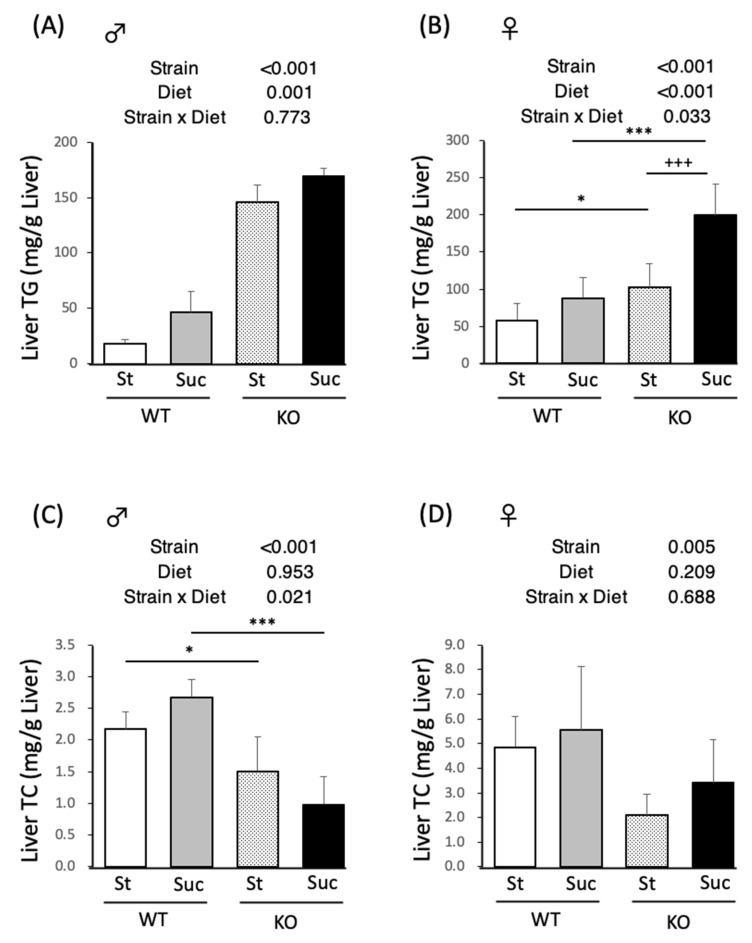
Liver TG and TC content according to diet. (**A**) Liver TG content in wild-type and PPARα^−/−^ male mice. (**B**) Liver TG content in wild-type and PPARα^−/−^ female mice. (**C**) Liver TC content in wild-type and PPARα^−/−^ male mice. (**D**) Liver TC content in wild-type and PPARα^−/−^ female mice. The inserted tables show the two-way ANOVA *p* values. * *p* < 0.05, *** *p* < 0.001 between mice fed the same diet. ^+++^
*p* < 0.001 between mice of the same strain. Data are expressed as the mean ± SEM, *n* = 6. WT, wild-type mice; KO, PPARα^−/−^ mice; St, starch diet; Suc, high-sucrose diet.

**Figure 2 biomedicines-10-02199-f002:**
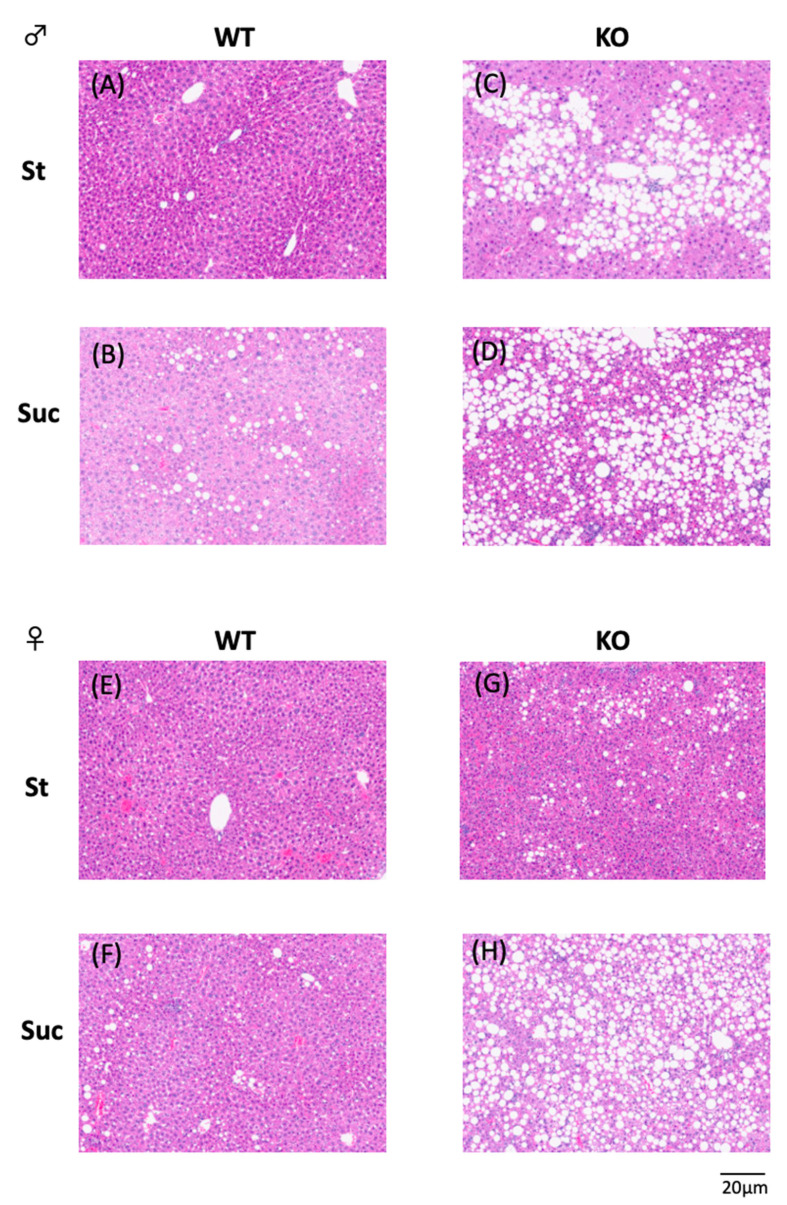
Hematoxylin and eosin (H & E)-stained liver sections from representative mice according to diet. (**A**) St-fed wild-type male mouse. (**B**) Suc-fed wild-type male mouse. (**C**) St-fed PPARα^−/−^ male mouse. (**D**) Suc-fed PPARα^−/−^ male mouse. (**E**) St-fed wild-type female mouse. (**F**) Suc-fed wild-type female mouse. (**G**) St-fed PPARα^−/−^ female mouse. (**H**) Suc-fed PPARα^−/−^ female mouse. WT, wild-type mice; KO, PPARα^−/−^ mice; St, starch diet; Suc, high-sucrose diet.

**Figure 3 biomedicines-10-02199-f003:**
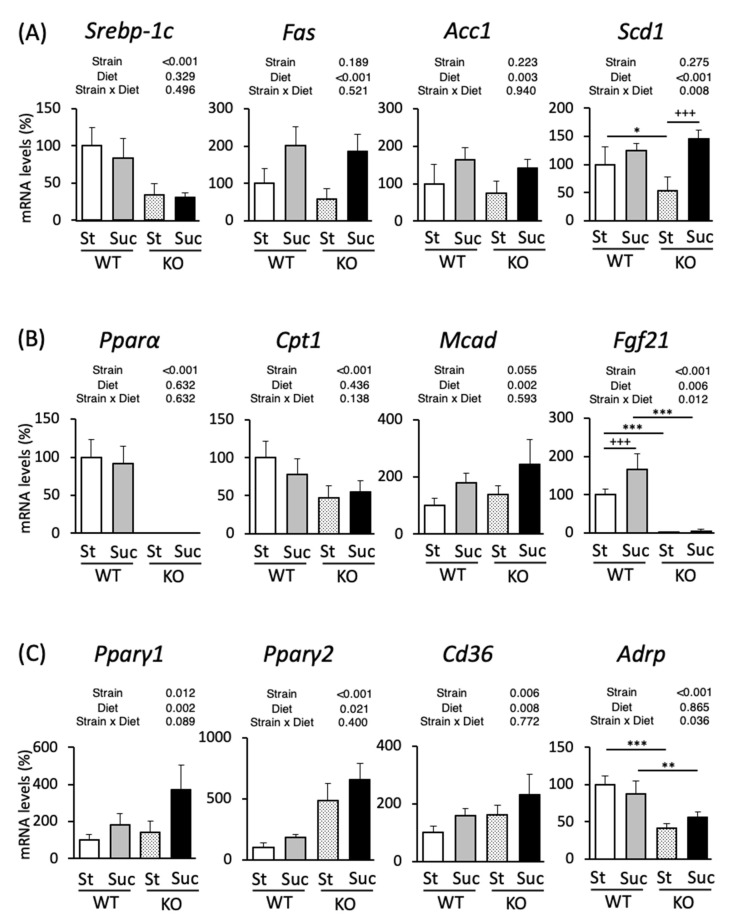
The hepatic gene expression in wild-type and PPARα^−/−^ male mice according to diet. Gene expressions related to fatty acid oxidation (**A**), fatty acid synthesis (**B**), and TG synthesis (**C**). The percentages of mRNA levels relative to St-fed WT mice are indicated. The inserted tables show the two-way ANOVA p values. * *p* < 0.05, ** *p* < 0.01, *** *p* < 0.001 between mice fed the same diet. ^+++^
*p* < 0.001 between mice of the same strain. Data are expressed as the mean ± SEM, *n* = 6. WT, wild-type mice; KO, PPARα^−/−^ mice; St, starch diet; Suc, high-sucrose diet.

**Figure 4 biomedicines-10-02199-f004:**
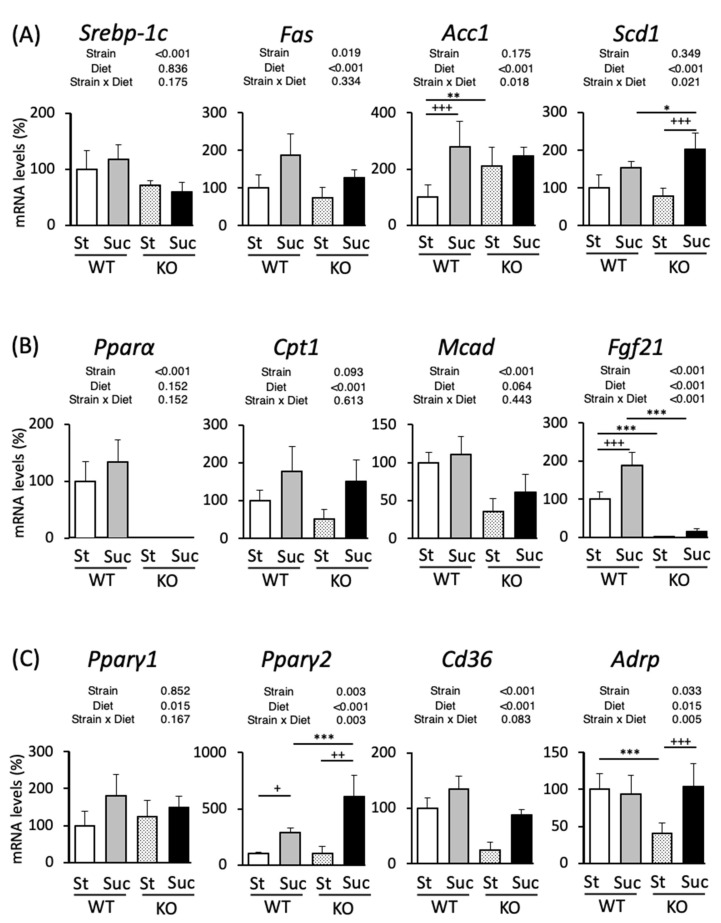
The hepatic gene expression in wild-type and PPARα^−/−^ female mice according to diet. The expression of genes related to fatty acid oxidation (**A**), fatty acid synthesis (**B**), and TG synthesis (**C**). The percentage of mRNA levels relative to St-fed WT mice are indicated. The inserted tables show the two-way ANOVA p values. * *p* < 0.05, ** *p* < 0.01, *** *p* < 0.001 between mice fed the same diet. ^+^
*p* < 0.05, ^++^
*p* < 0.01, ^+++^
*p* < 0.001 between mice of the same strain. Data are expressed as the mean ± SEM, *n* = 6. WT, wild-type mice; KO, PPARα^−/−^ mice; St, starch diet; Suc, high-sucrose diet.

**Figure 5 biomedicines-10-02199-f005:**
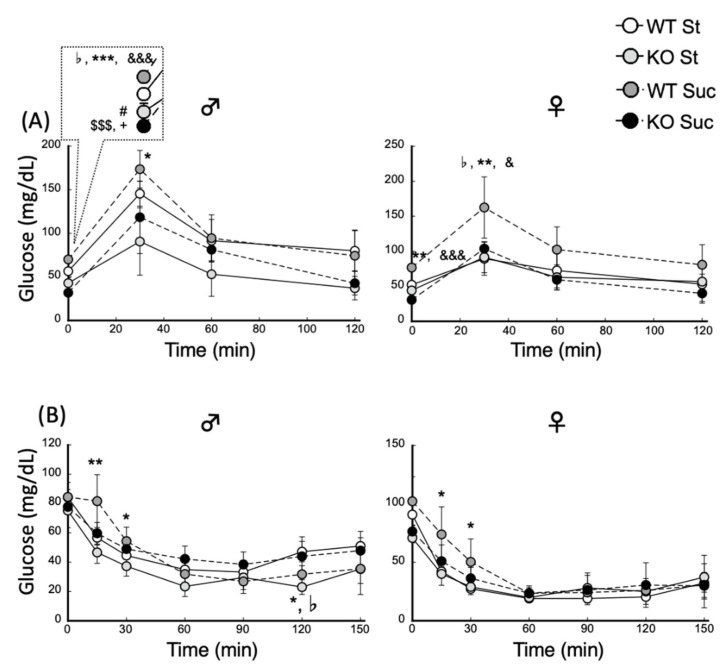
GTT and ITT according to diet in wild-type and PPARα^−/−^ mice. (**A**) Blood glucose concentrations during the GTT. (**B**) Blood glucose concentrations during the ITT. ^#^
*p* < 0.05 between Wt St and KO St, ^♭^
*p* < 0.05 between WT St and WT Suc; ^$$$^
*p* < 0.001 between Wt St and KO Suc; * *p* < 0.05, ** *p* < 0.01, *** *p* < 0.001 between WT Suc and KO St; ^+^
*p* < 0.05 between KO St and KO Suc, ^&^
*p* < 0.05, ^&&&^
*p* < 0.001 between Wt Suc and KO Suc. Data are expressed as the mean ± SEM, *n* = 6. WT, wild-type mice; KO, PPARα^−/−^ mice; St, starch diet; Suc, high-sucrose diet.

**Table 1 biomedicines-10-02199-t001:** Dietary composition.

Dietary Constituents	St	Suc
	g/100 g	
Safflower oil	1.0	1.0
Butter	3.6	3.6
Casein	19.7	19.7
α-Starch	66.3	16.5
Sucrose	-	49.8
Vitamin mix (AIN-93)	1.0	1.0
Mineral mix (AIN-93)	3.5	3.5
Cellulose powder	5.0	5.0
L-Cystine	0.3	0.3
	en%	
Fat	10	10
Carbohydrate	70	70
Protein	20	20

St, starch diet; Suc, high-sucrose diet; en%, energy%.

**Table 2 biomedicines-10-02199-t002:** Primers used for quantitative real-time PCR.

Gene	Forward Primer (5′ to 3′)	Reverse Primer (5′ to 3′)
*36b4*	GGCCCTGCACTCTCGCTTTC	TGCCAGGACGCGCTTGT
*Acc1*	GGACAGACTGATCGCAGAGAAAG	TGGAGAGCCCCACACACA
*Adrp*	AAGAGGCCAAACAAAAGAGCCAGGAGACCA	ACCCTGAATTTTCTGGTTGGCACTGTGCAT
*Cd36*	AATGGCACAGACGCAGCCT	GGTTGTCTGGATTCTGGA
*Cpt1*	GCACTGCAGCTCGCACATTACAA	CTCAGACAGTACCTCCTTCAGGAAA
*Fas*	GCTGCGGAAACTTCAGGAAAT	AGAGACGTGTCACTCCTGGACTT
*Fgf21*	ATGGAATGGATGAGATCTAGAGTTGG	TCTTGGTCGTCATCTGTGTAGAGG
*Mcad*	GATCGCAATGGGTGCTTTTGATAGAA	AGCTGATTGGCAATGTCTCCAGCAAA
*Pparα*	CCTCAGGGTACCACTACGGAGT	GCCGAATAGTTCGCCGAA
*Pparγ1*	GAGTGTGACGACAAGATTTG	GGTGGGCCAGAATGGCATCT
*Pparγ2*	TCTGGGAGATTCTCCTGTTGA	GGTGGGCCAGAATGGCATCT
*Scd1*	CCCCTGCGGATCTTCCTTAT	AGGGTCGGCGTGTGTTTCT
*Srebp-1c*	GGAGCCATGGATTGCACATT	CCTGTCTCACCCCCAGCATA

**Table 3 biomedicines-10-02199-t003:** Body and tissue weights and total energy intake.

	WT		KO		Two-Way ANOVA *p* Value		
	St	Suc	St	Suc	Strain	Diet	Strain × Diet
Male							
n	6	6	6	6			
Total Energy Intake (kcal/day/mouse)	16.6 ± 0.4	14.6 ± 0.4	15.7 ± 0.4	14.7 ± 1.3	0.271	0.001	0.225
Weight (g)							
BW at start	34.8 ± 2.0	34.9 ± 1.3	33.3 ± 1.3	34.9 ± 2.6	0.433	0.390	0.413
BW	36.0 ± 2.8	38.0 ± 1.6	35.7 ± 2.2	40.6 ± 5.4	0.503	0.055	0.397
Liver	1.300 ± 0.144	1.704 ± 0.137	1.746 ± 0.183	2.209 ± 0.345	<0.001	0.001	0.793
Epididymal WAT	1.388 ± 0.261	1.716 ± 0.107	1.267 ± 0.253	1.566 ± 0.468	0.381	0.054	0.924
Retroperitoneal WAT	0.311 ± 0.074	0.404 ± 0.057	0.207 ± 0.030	0.282 ± 0.098	0.005	0.028	0.795
Mesenteric WAT	0.493 ± 0.099	0.712 ± 0.071	0.461 ± 0.089	0.682 ± 0.271	0.698	0.012	0.998
Subcutaneous WAT	0.858 ± 0.216	0.996 ± 0.159	0.487 ± 0.054	0.711 ± 0.375	0.012	0.139	0.716
BAT	0.208 ± 0.039	0.209 ± 0.024	0.162 ± 0.014	0.191 ± 0.035	0.046	0.323	0.382
Quadriceps	0.294 ± 0.025	0.312 ± 0.016	0.289 ± 0.021	0.307 ± 0.051	0.755	0.247	0.998
Gastrocnemius	0.247 ± 0.023	0.241 ± 0.024	0.240 ± 0.018	0.267 ± 0.017	0.380	0.326	0.134
Female							
n	6	6	6	6			
Total Energy Intake (kcal/day/mouse)	15.7 ± 0.8	13.4 ± 1.1	14.3 ± 0.3	12.9 ± 0.4	0.009	<0.001	0.228
Weight (g)							
BW at start	26.4 ± 1.1	26.2 ± 1.0	26.0 ± 1.1	26.1 ± 1.0	0.621	0.935	0.750
BW	27.1 ± 0.8	26.5 ± 0.9	26.2 ± 1.6	26.6 ± 1.6	0.495	0.893	0.387
Liver	1.090 ± 0.128	1.251 ± 0.050	1.308 ± 0.110	1.469 ± 0.222	0.004	0.026	0.998
Periuterine WAT	0.715 ± 0.202	0.529 ± 0.261	1.008 ± 0.239	1.262 ± 0.273	<0.001	0.765	0.065
Retroperitoneal WAT	0.098 ± 0.017	0.085 ± 0.026	0.069 ± 0.020	0.071 ± 0.028	0.057	0.634	0.463
Mesenteric WAT	0.243 ± 0.043	0.255 ± 0.047	0.306 ± 0.065	0.363 ± 0.074	0.005	0.217	0.420
Subcutaneous WAT	0.381 ± 0.094	0.295 ± 0.051	0.317 ± 0.065	0.283 ± 0.036	0.230	0.063	0.409
BAT	0.109 ± 0.019	0.096 ± 0.009	0.107 ± 0.015	0.097 ± 0.011	0.905	0.097	0.874
Quadriceps	0.263 ± 0.022	0.283 ± 0.031	0.255 ± 0.023	0.258 ± 0.025	0.177	0.343	0.475
Gastrocnemius	0.212 ± 0.023	0.198 ± 0.012	0.192 ± 0.009	0.183 ± 0.010	0.016	0.106	0.729

WT, wild-type mice; KO, PPARα^−/−^ mice; St, starch diet; Suc, high-sucrose diet.

**Table 4 biomedicines-10-02199-t004:** Serum glucose, TG, TC and NEFA concentration.

	WT		KO		Two-Way ANOVA *p* Value		
	St	Suc	St	Suc	Strain	Diet	Strain × Diet
Male							
Glucose (mg/dL)	110.2 ± 6.9	114.2 ± 6.6	88.4 ± 6.1	94.8 ± 4.5	<0.001	0.107	0.698
TG (mg/dL)	123.5 ± 25.1	137.2 ± 20.6	120.8 ± 15.4	173.0 ± 32.3	0.191	0.015	0.132
TC (mg/dL)	97.0 ± 21.7	119.8 ± 46.7	96.5 ± 5.7	141.0 ± 26.2	0.487	0.034	0.465
NEFA (mEq/L)	0.481 ± 0.052	0.662 ± 0.134	0.533 ± 0.076	0.851 ± 0.140	0.039	<0.001	0.219
Female							
Glucose (mg/dL)	104.2 ± 5.0	107.3 ± 10.5	94.2 ± 7.8	94.5 ± 4.1	0.003	0.607	0.679
TG (mg/dL)	76.3 ± 10.6	76.0 ± 20.0	106.7 ± 19.2	80.3 ± 17.9	0.041	0.107	0.117
TC (mg/dL)	86.0 ± 10.2	97.9 ± 15.3	98.2 ± 18.9	102.3 ± 10.9	0.220	0.232	0.558
NEFA (mEq/L)	0.539 ± 0.143	0.500 ± 0.216	0.982 ± 0.231	1.112 ± 0.144	<0.001	0.601	0.339

WT, wild-type mice; KO, PPARα^−/−^ mice; St, starch diet; Suc, high-sucrose diet.

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
