# Peer review of "Peroxisome Proliferator-Activated Receptor α Has a Protective Effect on Fatty Liver Caused by Excessive Sucrose Intake"

_biomedicines, 2022, doi:10.3390/biomedicines10092199_

Round 1
Reviewer 1 Report
GENERAL COMMENTS
The topic is interesting as it provides important insight into the links between PPARalpha and NAFLD cuased by sucrose intake. Given the high prevalence of this pathology its clinical relevance is more than justified. The plentiful figures and tables are very welcome to summarize a huge amount of information.
The manuscript may benefit from considering the following aspects:
Abstract: it would be more informative to include some quantitative and statistical information.
Page 1, at the beginning of the Introduction: suggest to replace the first reference to to a more recent one like e.g. Francque SM, Marchesini G, Kautz A, et al. Non-alcoholic fatty liver disease: A patient guideline. JHEP Rep. 2021 Sep 17;3(5):100322.
The recent call to redefine NAFLD to metabolic associated fatty liver disease (MAFLD) focusing on obesity and metabolic dysfunction should be mentioned in the Introduction or in the Discussion.
Previous studies have shown that administration of fibroblast growth factor-19 (FGF-19) reverses diabetes, hepatic steatosis, hyperlipidemia, and adipose accretion in animal models of obesity. FGF-19 modulates hepatic fatty acid synthesis, a key process controlling glucose tolerance and triacylglycerol accumulation in liver, blood, and adipose tissue. Incubating primary hepatocyte cultures with recombinant FGF-19 suppresses the ability of insulin to stimulate fatty acid synthesis (ref. Bhatnagar S, Damron HA, Hillgartner FB. Fibroblast growth factor-19, a novel factor that inhibits hepatic fatty acid synthesis. J Biol Chem. 2009 Apr 10;284(15):10023-33). This effect was associated with a reduction in the expression of lipogenic enzymes. FGF-19 also suppresses the insulin-induced expression of sterol regulatory element-binding protein-1c (SREBP-1c), a key transcriptional activator of lipogenic genes. FGF-19 inhibition of lipogenic enzyme expression was not mediated by alterations in the activity of the insulin signal transduction pathway or changes in the activity of ERK, p38 MAPK, and AMP-activated protein kinase (AMPK). In contrast, FGF-19 increases the activity of STAT3, an inhibitor of SREBP-1c expression and decreases the expression of peroxisome proliferator-activated receptor-gamma coactivator-1beta (PGC-1beta), an activator of SREBP-1c activity. FGF-19 also increases the expression of small heterodimer partner (SHP), a transcriptional repressor that inhibits lipogenic enzyme expression via a SREBP-1c-independent mechanism. Inhibition of SREBP-1c activity by changes in STAT3 and PGC-1beta activity and inhibition of gene transcription by an elevation in SHP expression can explain the inhibition of lipogenesis caused by FGF-19 (ref. Bhatnagar S, Damron HA, Hillgartner FB. Fibroblast growth factor-19, a novel factor that inhibits hepatic fatty acid synthesis. J Biol Chem. 2009 Apr 10;284(15):10023-33). Moreover, in mice fed with a high-fat diet, palmitoleic acid supplementation stimulates the uptake of glucose in liver through activation of AMPK and FGF-21, dependent on PPARα (ref de Souza CO, Teixeira AAS, Biondo LA, Lima Junior EA, Batatinha HAP, Rosa Neto JC. Palmitoleic Acid Improves Metabolic Functions in Fatty Liver by PPARα-Dependent AMPK Activation. J Cell Physiol. 2017 Aug;232(8):2168-2177).
Given the relation between FGFs and obesity (refs Alvarez-Sola G, Uriarte I, Latasa MU, et al. Fibroblast growth factor 15/19 (FGF15/19) protects from diet-induced hepatic steatosis: development of an FGF19-based chimeric molecule to promote fatty liver regeneration. Gut. 2017 Oct;66(10):1818-1828 // Gómez-Ambrosi J, Gallego-Escuredo JM, Catalán V, et al. FGF19 and FGF21 serum concentrations in human obesity and type 2 diabetes behave differently after diet- or surgically-induced weight loss. Clin Nutr. 2017 Jun;36(3):861-868) the signalling pathways shared with SREBP-1c should be also mentioned.
Author Response
Responses for comments from the Reviewers in the documents:
Reviewer 1
- Abstract: it would be more informative to include some quantitative and statistical information.
Response: Thank you for your comments. We added the sentence that " PPARα-/- male and female mice fed a high-sucrose diet (Suc) showed 3.7- and 3.1- fold higher liver fat content than Suc-fed wild-type male and female mice, respectively." in Abstract.
- Page 1, at the beginning of the Introduction: suggest to replace the first reference to to a more recent one like e.g. Francque SM, Marchesini G, Kautz A, et al. Non-alcoholic fatty liver disease: A patient guideline. JHEP Rep. 2021 Sep 17;3(5):100322.
Response: Thank you for your comments. We replaced the first reference to the paper as you suggested.
- The recent call to redefine NAFLD to metabolic associated fatty liver disease (MAFLD) focusing on obesity and metabolic dysfunction should be mentioned in the Introduction or in the Discussion.
Response: Thank you for your comments. We added the sentence that " The term NAFLD was recently renamed metabolic dysfunction associated fatty liver disease (MAFLD) to better reflect the patient heterogeneity and pathogenesis [4] (Eslam, M.; Sanyal, A.J.; George, J.; International Consensus, P. MAFLD: A Consensus-Driven Proposed Nomenclature for Metabolic Associated Fatty Liver Disease. Gastroenterology 2020, 158, 1999–2014.e1. )." in Lines 35-37 in Introduction.
- Previous studies have shown that administration of fibroblast growth factor-19 (FGF-19) reverses diabetes, hepatic steatosis, hyperlipidemia, and adipose accretion in animal models of obesity. FGF-19 modulates hepatic fatty acid synthesis, a key process controlling glucose tolerance and triacylglycerol accumulation in liver, blood, and adipose tissue. Incubating primary hepatocyte cultures with recombinant FGF-19 suppresses the ability of insulin to stimulate fatty acid synthesis (ref. Bhatnagar S, Damron HA, Hillgartner FB. Fibroblast growth factor-19, a novel factor that inhibits hepatic fatty acid synthesis. J Biol Chem. 2009 Apr 10;284(15):10023-33). This effect was associated with a reduction in the expression of lipogenic enzymes. FGF-19 also suppresses the insulin-induced expression of sterol regulatory element-binding protein-1c (SREBP-1c), a key transcriptional activator of lipogenic genes. FGF-19 inhibition of lipogenic enzyme expression was not mediated by alterations in the activity of the insulin signal transduction pathway or changes in the activity of ERK, p38 MAPK, and AMP-activated protein kinase (AMPK). In contrast, FGF-19 increases the activity of STAT3, an inhibitor of SREBP-1c expression and decreases the expression of peroxisome proliferator-activated receptor-gamma coactivator-1beta (PGC-1beta), an activator of SREBP-1c activity. FGF-19 also increases the expression of small heterodimer partner (SHP), a transcriptional repressor that inhibits lipogenic enzyme expression via a SREBP-1c-independent mechanism. Inhibition of SREBP-1c activity by changes in STAT3 and PGC-1beta activity and inhibition of gene transcription by an elevation in SHP expression can explain the inhibition of lipogenesis caused by FGF-19 (ref. Bhatnagar S, Damron HA, Hillgartner FB. Fibroblast growth factor-19, a novel factor that inhibits hepatic fatty acid synthesis. J Biol Chem. 2009 Apr 10;284(15):10023-33). Moreover, in mice fed with a high-fat diet, palmitoleic acid supplementation stimulates the uptake of glucose in liver through activation of AMPK and FGF-21, dependent on PPARα (ref de Souza CO, Teixeira AAS, Biondo LA, Lima Junior EA, Batatinha HAP, Rosa Neto JC. Palmitoleic Acid Improves Metabolic Functions in Fatty Liver by PPARα-Dependent AMPK Activation. J Cell Physiol. 2017 Aug;232(8):2168-2177).
Given the relation between FGFs and obesity (refs Alvarez-Sola G, Uriarte I, Latasa MU, et al. Fibroblast growth factor 15/19 (FGF15/19) protects from diet-induced hepatic steatosis: development of an FGF19-based chimeric molecule to promote fatty liver regeneration. Gut. 2017 Oct;66(10):1818-1828 // Gómez-Ambrosi J, Gallego-Escuredo JM, Catalán V, et al. FGF19 and FGF21 serum concentrations in human obesity and type 2 diabetes behave differently after diet- or surgically-induced weight loss. Clin Nutr. 2017 Jun;36(3):861-868) the signalling pathways shared with SREBP-1c should be also mentioned.
Response: Thank you for your comments. FGF19/FGF15 is indeed an important factor. However, we did not analyze this factor, which is secreted from the small intestine upon stimulation from bile. We did not correct the small intestine sample. Also, we do not have system to measure FGF15 in serum. We will consider this factor at the next opportunity. As you pointed out, the relationship between FGF21 and SREBP-1c has already been reported: FGF21 suppresses SREBP-1c function. However, we did not observe the suppression of SREBP-1c by FGF21. So, we added the following sentence in the text in Lines 335-338 in Discussion. "On the other hand, it has been reported that FGF21 inhibits nuclear translocation of SREBP1c and decreases the amount of mature SREBP-1c protein [48]. However, the inhibition of SREBP-1c by FGF21 was not observed in the present study"
Reviewer 2 Report
This manuscript by Yamazaki et. al. investigated the molecular mechanisms related to fatty liver development in PPARα-deficient mice fed a high-sucrose diet. The SREBP-1c target gene expression was increased by sucrose intake, leading to fatty liver. Furthermore, PPARα-/- mice developed severe fatty liver. Thus, PPARα may work to prevent the development of fatty liver caused by excessive sucrose intake. The whole manuscript was well-organized, and the information provided in this study and the experimental methodology are interesting. However, the authors could have explained this manuscript more thoroughly. Overall, I recommend its publication after a minor revision with the following comments addressed.
1. Introduction
Are there other methods that should be discussed in the introduction as well, how does this approach compare to those? and what's the latest progress in this field?
And some other peroxidase-like materials (Journal of Materials Chemistry B 7 (16), 2613-2618); determination of glucose and cholesterol (Microchim. Acta 2019, 186, 269) can also be cited
2. Please improve the resolution of the figure.
3. The conclusion looks fine, and the main limitation also should be discussed as well.
Language issues,
4. There are a few areas where the English could be improved, such as some past and present tense.
5. There are some grammatical errors in this manuscript such as continuously forgetting to add ‘a’ or ‘the’ before a specific word which limits the clarity of the author’s writing.
Author Response
Responses for comments from the Reviewers in the documents:
Reviewer 2
- Introduction
Are there other methods that should be discussed in the introduction as well, how does this approach compare to those? and what's the latest progress in this field?
And some other peroxidase-like materials (Journal of Materials Chemistry B 7 (16), 2613-2618); determination of glucose and cholesterol (Microchim. Acta 2019, 186, 269) can also be cited
Response: Thank you for your comments. As you pointed out, we added the methods for measuring serum chemicals in Line 131-132 and References No. 34-36. The papers describing the method employed by the kit were added as a reference.
- Please improve the resolution of the figure.
Response: Thank you for your comments. The number indicating the p-value for the 2-way ANOVA was small, so we made the letters bigger. And we added the sentence that "The inserted tables show the two-way ANOVA p values." in Figure legends in Figures 1, 3 and 4.
- The conclusion looks fine, and the main limitation also should be discussed as well.
Response: Thank you for your comments. We added the sentence that "More detailed analysis may be possible by using postmenopausal or ovariectomized female PPARα-/- mice." in Lines 373-374 in Discussion.
- There are a few areas where the English could be improved, such as some past and present tense.
- There are some grammatical errors in this manuscript such as continuously forgetting to add ‘a’ or ‘the’ before a specific word which limits the clarity of the author’s writing.
Response: Based on your comments regarding the readability of the text, the manuscript has been rechecked by a professional editor who is a native speaker of English. We attached the native check certificate.
